# The *Arabidopsis* transcription factor ABIG1 relays ABA signaled growth inhibition and drought induced senescence

Tie Liu, Adam D Longhurst, Franklin Talavera-Rauh, Samuel A Hokin, M Kathryn Barton*

Department of Plant Biology, Carnegie Institution for Science, Stanford, United States

**Abstract** Drought inhibits plant growth and can also induce premature senescence. Here we identify a transcription factor, ABA INSENSITIVE GROWTH 1 (ABIG1) required for abscisic acid (ABA) mediated growth inhibition, but not for stomatal closure. *ABIG1* mRNA levels are increased both in response to drought and in response to ABA treatment. When treated with ABA, *abig1* mutants remain greener and produce more leaves than comparable wild-type plants. When challenged with drought, *abig1* mutants have fewer yellow, senesced leaves than wild-type. Induction of ABIG1 transcription mimics ABA treatment and regulates a set of genes implicated in stress responses. We propose a model in which drought acts through ABA to increase *ABIG1* transcription which in turn restricts new shoot growth and promotes leaf senescence. The results have implications for plant breeding: the existence of a mutant that is both ABA resistant and drought resistant points to new strategies for isolating drought resistant genetic varieties.

*For correspondence: kbarton@CarnegieScience.edu

**Competing interests:** The authors declare that no competing interests exist.

## Introduction

One way for plants to withstand drought is to slow, or stop, new growth thereby conserving resources until better times return (*Dolferus, 2014*). Growth arrest coincides with predictable dry periods during the life cycle – such as in seed development or seasonal bud dormancy. Growth arrest or reduction can also occur in response to sporadic drought (e.g. *Marc and Palmer, 1976*; *Verelst et al., 2013*) More extreme than growth arrest is a response in which parts of the plant are sacrificed, as for instance in drought induced leaf senescence, allowing the plant to recover nutrients and to reduce the water cost of maintaining the leaf (*Munné-Bosch et al., 2004*).

Among plant hormones, abscisic acid (ABA) is the best known for its ability to allow plants to withstand drought. In response to desiccation, levels of the ABA biosynthetic enzyme AtNCED3 in the vascular parenchyma increase in Arabidopsis (*Endo et al., 2008*). This causes an increase in ABA which causes plants to close stomatal apertures, thus reducing water loss (*Schroeder et al., 2001*). Increases in ABA concentration during seed development impose maturation and dormancy on seeds (*Finkelstein, 2013*). ABA acts by binding to PYR-like co-receptors and bringing them together with PP2C family phosphatases (*Cutler et al., 2010*). This frees downstream SNRK2 kinases from repression by the PP2C phosphatase, allowing the kinase to modify protein targets at the plasma membrane - to alter turgor in guard cells - or to modify a set of transcription factors - to promote maturation and dormancy in the developing seed.

The role of ABA in controlling vegetative growth is less clear. Exogenously applied ABA slows growth of roots and shoots: it promotes dormancy in axillary buds (*Shimizu-Sato and Mori, 2001*),

inhibits formation and growth of leaves (*Marc and Palmer, 1976*; *Verelst et al., 2013*) and inhibits root growth (*DeSmet et al., 2003*). ABA also can promote drought induced leaf senescence (*Munné-Bosch and Alegre, 2004*).

Paradoxically, ABA appears to be required for growth as well. This conclusion is based on the observation that biosynthetic mutants with reduced levels of ABA are smaller than wild type plants (*Barrero et al., 2005*). Supplementation with ABA restores growth of plants grown under well watered conditions indicating the growth defect is not a secondary consequence of water loss. Taken at face value, these observations indicate that ABA promotes growth as well as inhibits it.

In this paper, we describe a transcription factor that is required for inhibition of growth in response to ABA. We propose that this factor does not act on the canonical ABA events of stomatal closure and seed dormancy but rather in a branch through which environmental stresses act through ABA to restrict growth.

## Results

A tenet of systems biology is that agents inhabiting the same node of a regulatory network share a biological function. REVOLUTA (REV) and KANADI1 (KAN1), opposing regulators of leaf polarity and meristem formation, also oppositely control components of the ABA signaling pathway (*Reinhart et al., 2013*). A connection between the ad/abaxial developmental and ABA signaling networks had not been predicted based on mutant phenotypes, and the biological relevance of the connection is not understood. Nevertheless, since ABA signaling genes are prominent among the small set of *ORK* genes (genes <u>o</u>ppositely <u>r</u>egulated by the REV and KAN transcription factors (*Reinhart et al., 2013*; *Figure 1A*), we reasoned that other genes oppositely regulated by them may also play a role in ABA signaling in the plant.

The set of eight *ORK* genes (<u>OPPOSITELY REGULATED BY REVOLUTA AND KANADI</u>) includes two genes involved in ABA signaling: *PYL6* and *CIPK12*. *PYL6* encodes a member of the family of ABA receptors (*Park et al., 2009*). *CIPK12* encodes a member of a family of SNRK3 kinases that play a role in the ABA regulatory pathway (*Qin et al., 2008*; *Lumba et al., 2014*). Except for *ZFP8*, which plays a role in trichome development (*Gan et al., 2007*), the function of the other five genes at this regulatory node is unknown.

*HOMEOBOX FROM ARABIDOPSIS THALIANA 22* (AT4G37790/ HAT22, *Figure 1B*) was chosen for study because it shows rapid and robust opposite regulation by REV and KAN1, because it is highly conserved among land plants and because its function in the plant is largely unknown. *HAT22* is one of 10 *Class II HOMEODOMAIN-LEUCINE ZIPPER (HD-ZIPII)* genes (*Ciarbelli et al., 2008*). We propose to rename this gene <u>A</u>BA <u>I</u>NSENSITIVE <u>G</u>ROWTH 1 (ABIG1) to better reflect its function in the plant (see below).

### Expression of *ABIG1/HAT22* mRNA increases with ABA and drought

*ABIG1* mRNA levels increase in liquid grown seedlings treated with ABA (*Figure 1C and D*). *ABIG1* mRNA levels did not increase in homozygous *abig1-1* mutants treated with ABA (*Figure 1C*), presumably because the inserted DS element causes transcript termination early in the *ABIG1* transcript. The increase in *ABIG1* mRNA is reduced in ABA treated *abi1* mutant seedlings defective for the ABI1 PP2C protein phosphatase, a co-receptor for ABA (p (treatment by genotype) = 0.0033, 2 way ANOVA; *Figure 1D*). Thus, ABA stimulation of *ABIG1* mRNA increase utilizes, at least in part, the core ABA signaling pathway.

Drought increases ABA levels. If *ABIG1* responds to endogenous ABA, we would expect *ABIG1* mRNA levels to increase in plants from which water has been withheld. Indeed, *ABIG1* mRNA levels increase with decreasing soil moisture (*Figure 1E*). The response of *ABIG1* expression to drought has also been noted by *Su et al. (2013)* who found *ABIG1/HAT22* mRNA up-regulated in flowers in response to drought.

The *abig1-1/ GT7363* line carries an engineered DS transposable element inserted into the sixth codon of the *ABIG1* coding sequence (*Figure 1B*). This DS element carries a *beta-GLUCURONIDASE (GUS)* gene (*Springer et al., 2000*) that is expressed as part of the disrupted *ABIG1* transcript. In *abig1-1/+* plants, *GUS* is expressed in cells surrounding the vascular strand with highest levels in the petioles, the hypocotyl and subtending the shoot apical meristem and youngest leaf primordia (*Figure 1G–I*). Cross sections show GUS expression in the petiole to be highest in the vascular

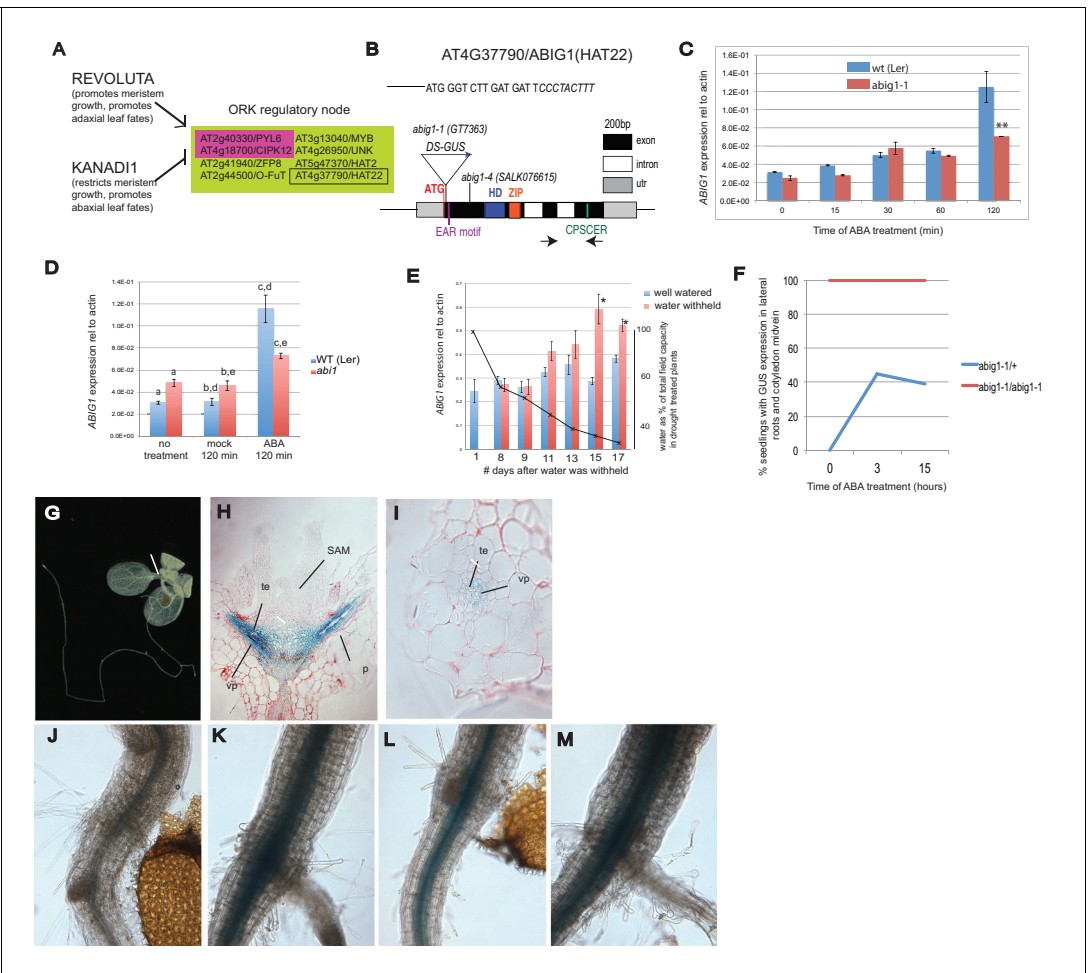

**Figure 1.** Control of Arabidopsis Abscisic Acid Insensitive Growth 1 (ABIG1/HAT22) expression. (**A**) Set of eight *ORK* genes (*OPPOSITELY REGULATED BY REVOLUTA AND KANADI*). (**B**) Structure of the *ABIG1* gene. HD - DNA binding homeodomain, ZIP-leucine zipper responsible for homodimerization, EAR – domain responsible for interaction with TOPLESS family co-repressors, CPSCER – conserved domain of unknown function. Arrows - position of PCR primers for *ABIG1* Q-RTPCR in panels C-E. Position of gene trap insertion is shown. The sequence above indicates junction between *ABIG1* gene and DS insertion (italics). (**C**) *ABIG1* mRNA in wild type and *abig1-1* homozygous mutant plants in response to ABA treatment (whole plants grown in liquid media). **p<0.01. (**D**) *ABIG1* mRNA in wild type and *abscisic acid insensitive (abi1-1)* mutant plants in response to ABA treatment (whole plants grown in liquid media). Letters indicate pairs showing significant differences (p<0.05; t-test). (**E**) *ABIG1* mRNA in response to drought treatment. Water was withheld from rosette stage plants beginning on day one. RNA is from stems and leaves of plants grown in soil. Asterisk indicates time points where watered and unwatered plants differ significantly (p<0.05; t test). (In all cases, bars indicate ± s.e.m. for three biological replicates. The data in each graph are from independent experiments.) (**F**). Increase in extent of GUS expression in *abig-1/+* plants treated with ABA. (**G**) GUS expression from GT7363 genetrap in *abig1-1/+* plant. White line points to area of expression at basal petiole and subtending the shoot apex. (**H**) Longitudinal thin section through *abig1-1/+* seedling showing GUS expression along vascular strands. SAM – shoot apical meristem and associated young leaf primordia; vp – vascular parenchyma; te – tracheary elements; p – petiole. (**I**) Cross section through a leaf petiole of an *abig1-1/+* seedling showing GUS expression in vascular parenchyma cells associated with tracheary elements of xylem. (**J**) GUS stained hypocotyl/root junction of *abig1-1/+* seedling. (**K**) GUS stained hypocotyl/root junction of *abig1-1/+* seedling treated with ABA for three hours. (**L**) GUS stained hypocotyl/root junction of *abig1-1/abig1-1* seedling. (**M**) GUS stained hypocotyl/root junction of *abig1-1/abig1-1* seedling treated with ABA for three hours.

The following figure supplement is available for figure 1:

**Figure supplement 1.** *ABIG1* mRNA levels are reduced in homozygous *abig1-4* mutants.

parenchyma associated with the tracheary elements of the xylem. It is noteworthy that xylem is adaxial to the phloem. This may explain the observed regulation of *ABIG1* transcription by the REVOLUTA and KANADI regulators of ad/abaxial leaf polarity.

Consistent with the Q-RTPCR measurements above, the transfer of heterozygous *abig1/+* plants to solid media containing 5 microMolar ABA resulted in an increase in the intensity and extent of GUS expression (*Figure 1F,K*). Expression extended radially in the hypocotyl and further up the cotyledon midvein. These patterns of expression are consistent with observations showing exogenously added ABA traveling upward from the root toward the shoot (*Waadt et al., 2014*). The location of *ABIG1/HAT22* mRNA expression in the vascular parenchyma is also consistent with the location of increased ABA biosynthetic enzymes in response to drought (*Endo et al., 2008*).

HD-ZIPII proteins are known to bind to their own promoters and negatively regulate their own transcription (*Ciarbelli et al., 2008*; *Turchi et al., 2013*). Indeed, homozygous *abig1-1* plants show higher and more expanded GUS expression than heterozygotes (compare *Figure 1J* to *Figure 1L*). ABA treatment did not increase the already high levels of reporter expression seen in *abig1-1* homozygotes. (*Figure 1L,M*). This suggests that ABA acts indirectly to increase *ABIG1* levels by interfering with the negative autoregulatory action of the ABIG1 protein. This is an attractive hypothesis given the absence of ABREs (ABA Response elements) in the 1000 bp upstream of *ABIG1*.

## *abig1/hat22* mutants display normal stomatal closure and germination responses

*abig1-1* mutant seeds and plants do not behave the same as canonical ABA resistant mutants. In contrast to canonical ABA insensitive (abi) mutants (*Koorneef et al., 1984*; *Finkelstein, 1994*), *abig1-1* mutants were similar to wild type seeds in their inability to germinate in the presence of ABA, a potent germination inhibitor (*Figure 2A*). Another characteristic of ABA signaling defective mutants is a failure of stomatal closure in response to ABA application. ABA applied to the epidermis of wild-type and *abig1-1* mutant leaves caused similar decreases in stomatal apertures (*Figure 2B*).

## *abig1/hat22* mutants are resistant to ABA mediated growth inhibition of shoot tissues

To determine if *ABIG1* affects responses to ABA after germination, we transferred seedlings at seven days from standard MS medium to MS medium supplemented with ABA. Under this treatment regimen ABA inhibited growth and caused leaf yellowing in wild type seedlings (*Figure 3*). ABA caused fewer leaves to develop at the shoot apex, and caused leaves to have shorter petioles and smaller leaf blades. *abig1-1* mutants showed less response to ABA than wild type for leaf but not root growth: *abig1* mutants differed in their response to ABA with regard to leaf number (p(genotype by [ABA]) <0.0001) and petiole length (p(genotype by [ABA]) <0.02) but not root length (p(genotype by [ABA]) = 0.285) as determined using a 2 way ANOVA analysis. We conclude that the wild type

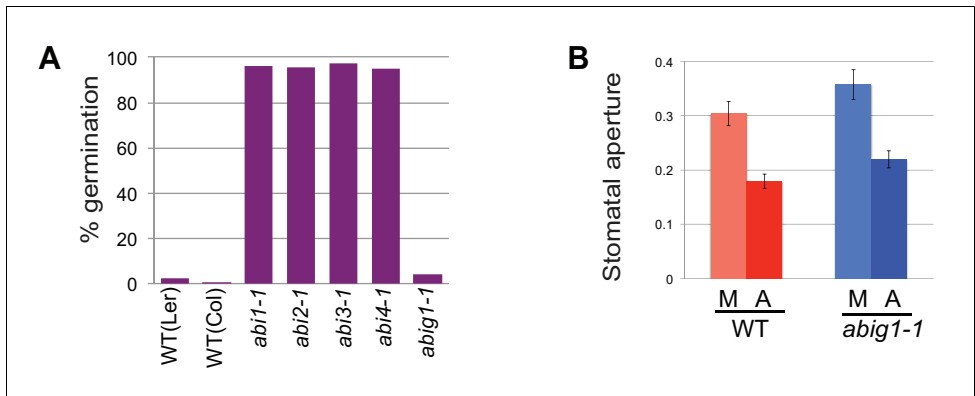

**Figure 2.** Stomatal closure and germination of abig1-1 mutant seedlings in response to exogenous ABA. (**A**) Germination of wild type (wt) and mutant seeds on MS medium with 5 microMolar ABA. Germination of *abi1-1*, *abi2-1*, *abi3-1* and *abi4-1* mutants is ABA resistant while germination of wild type and *abig1-1* seedlings is ABA sensitive. (**B**) Stomatal closure induced by ABA (A) or mock (M) treatment is similar in wild type and *abig1-1* mutant plants.

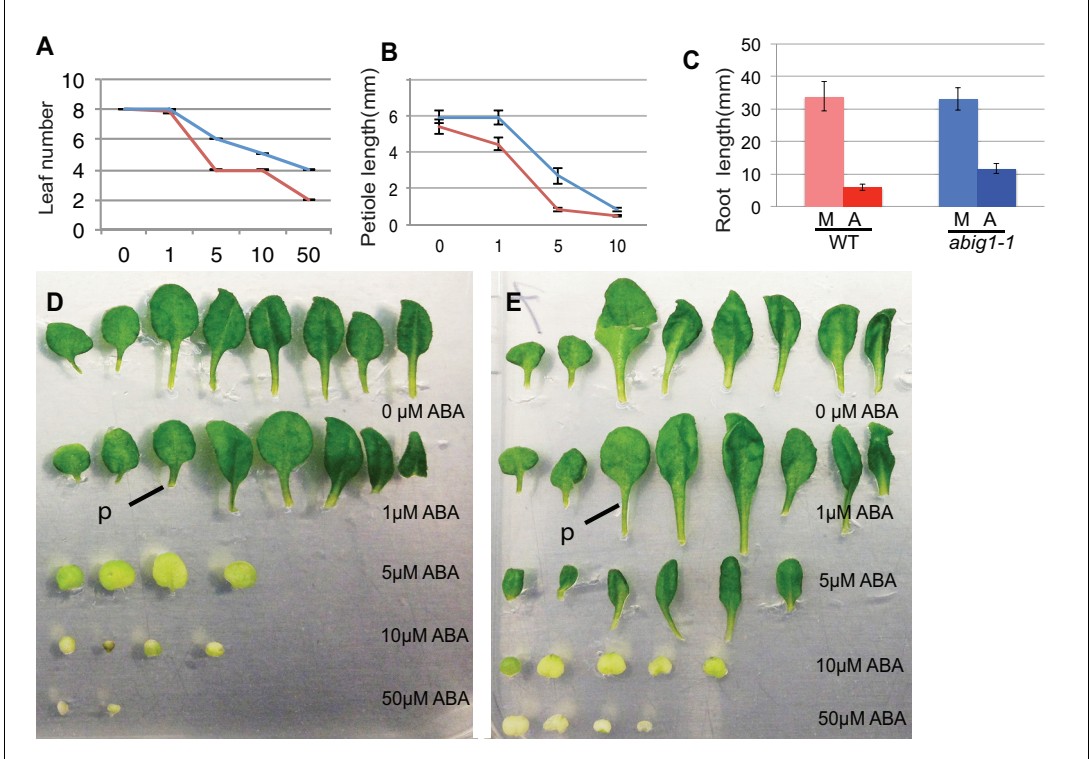

**Figure 3.** Vegetative growth of *abig1-1* mutant seedlings in response to exogenous ABA. One week old wt and *abig1-1* mutant plants were transferred to the ABA containing medium of varying concentration (x axis is in microMolar). Leaf number (**A**), petiole length of leaf three (**B**) and primary root length (**C**) were measured after 14 days on ABA. Red – wild type Ler; blue – *abig1-1*. Representative wild type (**D**) and *abig1* plants (**E**) grown on increasing concentration of ABA. Each row of leaves is from a single plant with the first formed leaf to the left. p = petiole of leaf three.

The following figure supplement is available for figure 3:

**Figure supplement 1.** Response of *abig1-4* mutants to exogenous ABA.

*ABIG1* function is required for ABA-mediated inhibition of leaf production and petiole growth. These experiments were repeated with a second loss of function allele, *abig1-4*, and similar results were obtained (*Figure 1—figure supplement 1*; *Figure 3—figure supplement 1*).

## Overexpression of ABIG1/HAT22 mimics ABA application

Induced overexpression of *ABIG1* (achieved by placing the *ABIG1* cDNA under the control of an estradiol inducible *XVE* transcription factor [*Zuo et al., 2000*]) mimicked application of ABA to wild type plants. Leaves developed with small blades and short petioles, leaves were yellow, and fewer leaves were made (*Figure 4*). Thus, increased *ABIG1* mRNA levels are sufficient to cause growth inhibition and leaf yellowing. We note that this is consistent with results by *Köllmer et al. (2011)* who found a modest increase in leaf senescence when *ABIG1 (HAT22)* was expressed under control of the 35S promoter. The more severe yellowing and seedling arrest phenotype seen in our experiments may be because the conditional nature of the estradiol induced promoter makes it possible for us to recover transgenic plants with high expression levels.

In summary, examination of loss and gain of function mutants in *ABIG1* support the hypothesis that this transcription factor contributes to ABA induced growth inhibition and leaf yellowing in the plant, and is sufficient to mimic the effect of exogenous ABA on growth inhibition and leaf yellowing. Such behavior is consistent with a model in which ABA inhibits growth of the shoot by causing increased levels of ABIG1 expression. Addition of increasing concentrations of ABA to estradiol treated *XVE:ABIG1* plants did not make the phenotype more extreme than when estradiol alone was

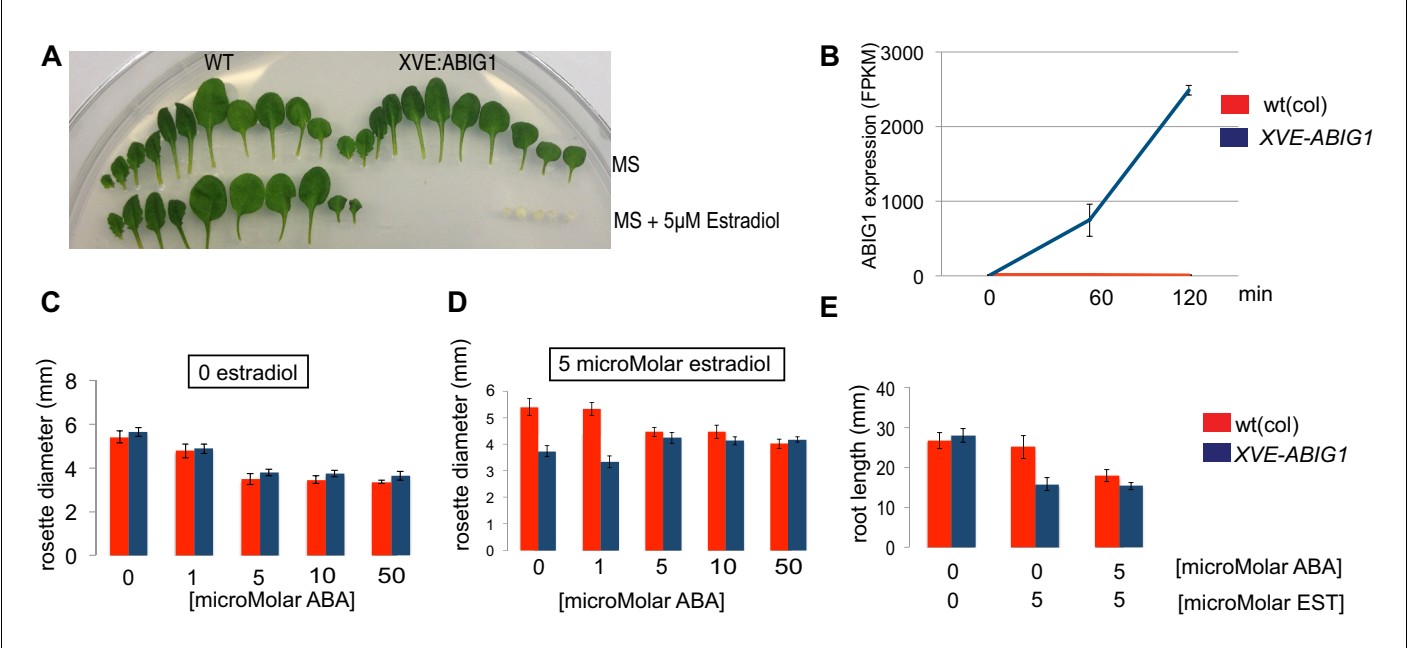

**Figure 4.** Induced overexpression of ABIG1 mimics ABA treatment. (**A**) Wild type plants (left) grown on standard MS plates (upper row) and on plates containing 5 microMolar estradiol (lower row). Each row is a series of leaves taken from a single rosette (first made leaves are to the left). *XVE: ABIG1* plants (right) grown on standard MS plates (upper row) and on plates containing 5 microMolar estradiol (lower row). (**B**). ABIG1 mRNA levels following estrogen induction of XVE:ABIG1. (**C**) Rosette diameter in wild type (red) and *XVE: ABIG1* plants (blue) treated with increasing concentrations of ABA. (**D**) Rosette diameter in wild type and *XVE: ABIG1* plants treated with 5 microMolar estradiol and with increasing concentrations of ABA. (**E**) Length of the primary root in wild type and *XVE:ABIG1* plants treated with 5 microMolar estradiol and with 5 microMolar ABA.

added (*Figure 4C–E*). This supports a model in which the major role for ABA in regulating plant growth is through its role in promoting the *ABIG1* transcript accumulation.

## ABIG1/HAT22 mutants show less leaf yellowing in response to drought

The finding that *ABIG1* mRNA levels increase in response to both ABA and drought and that *ABIG1* is necessary for ABA mediated growth inhibition and leaf yellowing suggested that *ABIG1* might play a role in response to drought. To test this, plants were grown two per pot with one wild type and one mutant in each pot. At 34 days post-germination, just prior to bolting, the pots were split into two groups. The first group was well watered for the duration of the experiment while water was withheld from the second group. Under well-watered conditions, *abig1-1* mutants had a lower percentage of yellow, senesced leaves than wild-type (*Figure 5*). For one experimental replicate, the amount of chlorophyll present in leaves was measured (*Figure 5G*). It was found to decrease significantly more in the wild type than in the mutant.

After 17 days of withholding water, this difference became significantly more pronounced with the percent yellow or senesced leaves more than doubling in the wild type but only increasing modestly in the homozygous *abig1-1* mutant (p(treatment × genotype) = 0.0008; *Figure 5B,D*). Under water withheld conditions, wild type plants made fewer side branches than under well watered conditions while *abig1* mutants continued to produce the same number of side branches (p(treatment × genotype) = 0.0114; *Figure 5E*). The wild type plants were unable to remain upright while the mutant plants remained erect (*Figure 5B* ; 100% erect mutant plants (n = 9) vs 37.5% erect wild type plants (n = 9) in one experiment with similar results in 3 additional replicated experiments). Wild type plants also showed less extensive root systems than the *abig1-1* mutants (*Figure 5C*). This experiment was repeated multiple (greater than three) times with similar results.

Similar results were obtained with the *abig1-4* allele except that the *abig1-4* mutants made the same number of side branches as wild type under dry conditions (*Figure 5—figure supplement 1*). Since both the Ds insertion in *abig1-1* and the T-DNA insertion in *abig1-4* disrupt the gene close to

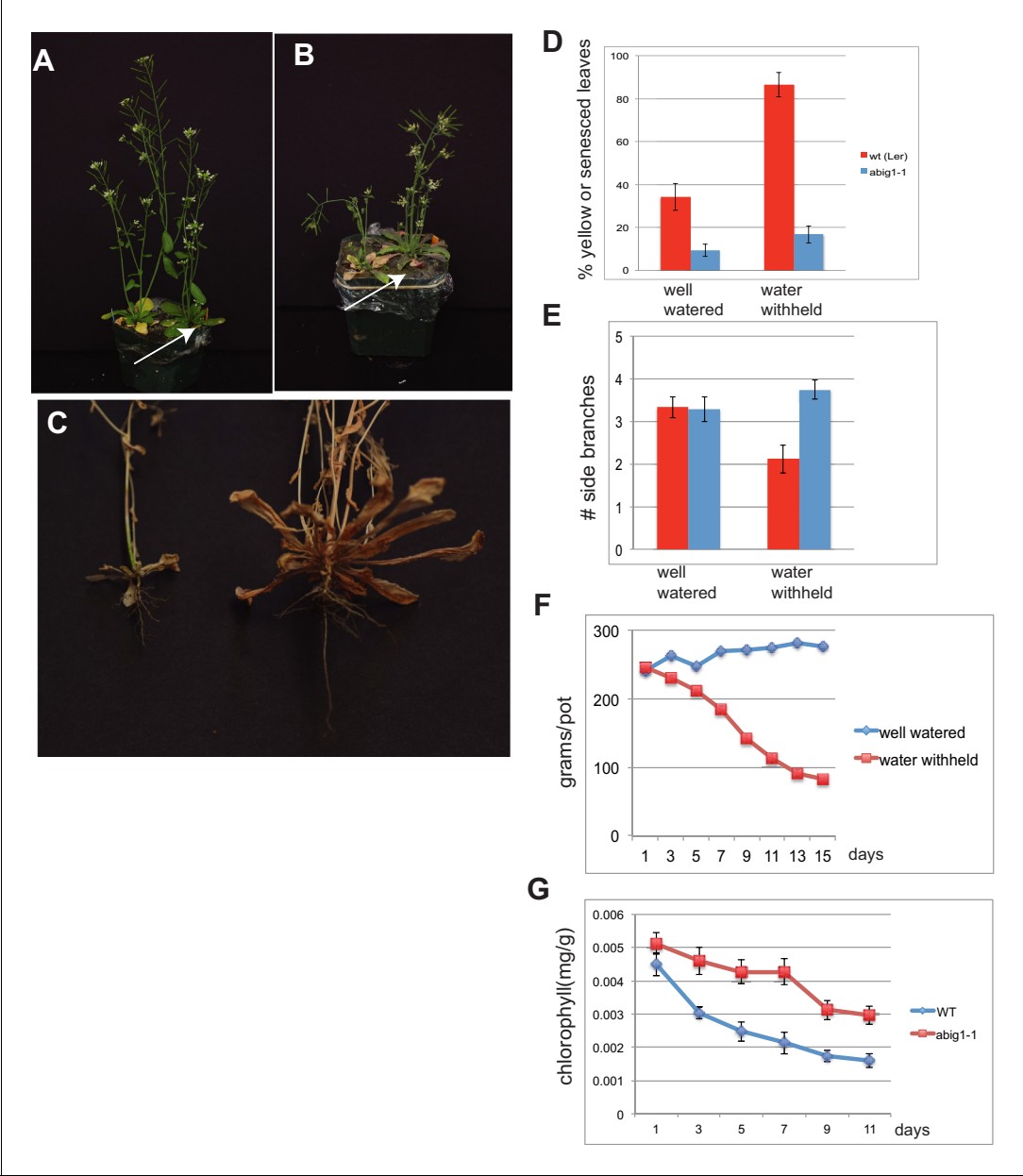

**Figure 5.** abig1-1 mutants are resistant to drought. (**A**) Well watered plants. Wild-type Ler (left) and *abig1-1* (right) plants grown in a shared pot. (**B**) Plants from which water was withheld for 17 days. Wild-type (left) and *abig1-1* (right) plants grown in a shared pot. Note yellow, senesced rosette leaves and drooping of wild type plant. (**C**) Root systems of wild type (left) and *abig1-1* (right) plants from which water was withheld. Plants were harvested at the end of their lifecycle at which point both genotypes are fully senesced. (**D**) Percentage of leaves that had turned yellow after 17 days of withheld water. Total leaves averaged 9 for wild-type and 10 for mutant plants. (**E**) Number of branches formed in plants from which water was withheld. A branch was scored as present if longer than 1 cm. (**F**) Rate of water loss from pots during the water withholding period. (**G**) Chlorophyll content of leaves number 3–8 in drought treated wt and *abig1-1* mutant plants over the course of the experiment. This experiment (10 pairs of mutant and Ler wild-type) is a replicate of the experiment shown in **A–E** (16 pairs of mutant and Ler wild-type).

The following figure supplements are available for figure 5:

**Figure supplement 1.** Response of *abig1-4* mutants to drought treatment.

**Figure supplement 2.** Measurement of levels of *SAG113/HAI1* mRNA levels in wild type and *abig1* mutants in response to exogenously added ABA.

**Figure supplement 3.** Response of *abig1-1* mutant outcrossed to Columbia to drought.

its 5 prime end, this phenotypic difference is unlikely due to a difference in allele strength but rather to differences in genetic background. In support of this, the phenotype of the *abig1-1* mutation is similar to that of *abig1-4* when it is outcrossed to wild type Columbia, the background which *abig1-4* is in (*Figure 5—figure supplement 3*). We conclude that the *ABIG1* transcription factor accelerates leaf senescence and acts to prevent the plant from remaining erect under dry conditions. This result was unexpected since ABA resistant mutants isolated in the past have been associated with drought sensitivity rather than drought resistance (see e.g. *Wang et al., 2009*).

## Genes regulated by ABIG1 are enriched for stress related genes including ABA and ethylene response loci

To determine what genes the ABIG1 transcription factor controls, we treated *XVE:ABIG1* plants and wild-type plants with 5 microMolar estradiol for 0, 60 and 120 min. Plants were flash frozen at each time point and RNA was isolated and subjected to RNA-seq. This treatment resulted in robust up-regulation of *ABIG1* mRNA (*Figure 4B*; *Supplementary file 1*).

Using the CUFFDIFF application (*Trapnell et al., 2013*) we first identified genes differentially expressed after estradiol treatment (p adjusted <0.01; |FC|>1.5 (log base 2)). We then removed any genes that showed a (time of treatment by genotype) interaction p value greater than 0.05 (Two way-ANOVA). This limits the set to genes that respond differently in wild-type vs. *XVE:ABIG1*. We also removed genes for which one or more time points resulted in a 'no test' call by CUFFDIFF. The final list includes 18 up-regulated mRNAs (*Figure 6A* and *Supplementary file 1*) and 14 down-regulated mRNAs (*Figure 6B* and *Supplementary file 1*).

Down-regulated, but not up-regulated, genes, showed statistically significant over-representation for Gene Ontology (GO) terms associated with response to stress, response to abiotic or biotic stimulus, and signal transduction (*Figure 6C*). The interaction of the EAR domain of *ABIG1* with the TOP-LESS co-repressor (*Causier et al., 2012*; T. Liu and M.K.Barton, unpublished), and the demonstrated ability of HD-ZIPII factors to repress their own transcription (*Ohgishi et al., 2001*; *Ciarbelli et al., 2008*) are evidence that *ABIG1* acts a repressor. Direct targets of *ABIG1* are therefore expected to fall among the genes down-regulated by *ABIG1* induction.

The promoters (500 bp upstream of the ATG) of the down regulated genes are enriched for bHLH, MADS and a homeodomain transcription factor binding sites (p=1.6E-4, p=1.5E-2 and p=1.9E-2 respectively; P-Scan (*Zambelli et al., 2009*)). The strong representation of additional types of transcription factor binding sites, especially bHLH binding sites, within the down-regulated genes points to combinatorial regulation of these loci by *ABIG1* and as yet unspecified bHLH proteins.

Promoters for up-regulated genes were enriched for bZIP transcription factor binding sites (p = 4.1E-3).

Notable among the regulated genes are: *CYP707A3* which encodes an enzyme that catabolizes ABA (*Umezawa et al., 2006*); *GRF5* which regulates leaf senescence and chloroplast number (*Vercruyssen et al., 2015*); *BFN1*, a gene encoding a nuclease involved in leaf and stem senescence (Farage-Barhom [remove (Farage-Barhorn] (*Farage-Barhom et al., 2011*); *ATHB5*, a gene implicated in drought resistance (*Johannesson et al., 2003*)and *TOPLESS RELATED 3 (TPR3)*, a member of the *TOPLESS* co-repressor gene family and a likely partner of the ABIG1 transcription factor (*Causier et al., 2012*; *Long et al., 2006*).

Because several plant hormones influence ABA sensitivity and drought tolerance we queried the dataset with gene lists for genes involved in biosynthesis and/or signaling for the plant hormones abscisic acid, ethylene, jasmonate and cytokinin. These hormones were chosen because they have been implicated in promoting (ABA, jasmonate and ethylene) or preventing (cytokinin) senescence (*Kim et al., 2015*; *Pré et al., 2008*; *Buchanan-Wollaston, 1996*). We also included a set of genes associated with chlorophyll degradation. Six genes passed this filter (*Supplementary file 1*; *Figure 6D*). In addition to the ABA catabolic enzyme gene, CYP707A2, induction of ABIG1 also regulates At2g17820/HK1, a gene encoding a receptor that mediates ABA, drought and salt stress responses (*Wollbach et al., 2008*) and the gene encoding the ethylene receptor ERS2. Finally, two transcription factor genes in the jasmonate pathway, *ORA59* and *TIFY7* were down-regulated in this experiment. The identities of these regulated genes support a model in which ABIG1 promotes drought-induced senescence in part by changing the amount of ABA in the plant while also altering the sensitivity of the cell to ABA, ethylene and jasmonate. While all of these changes occur rapidly (within hours) after the induction of ABIG1 mRNA synthesis, the effects may nonetheless be indirect

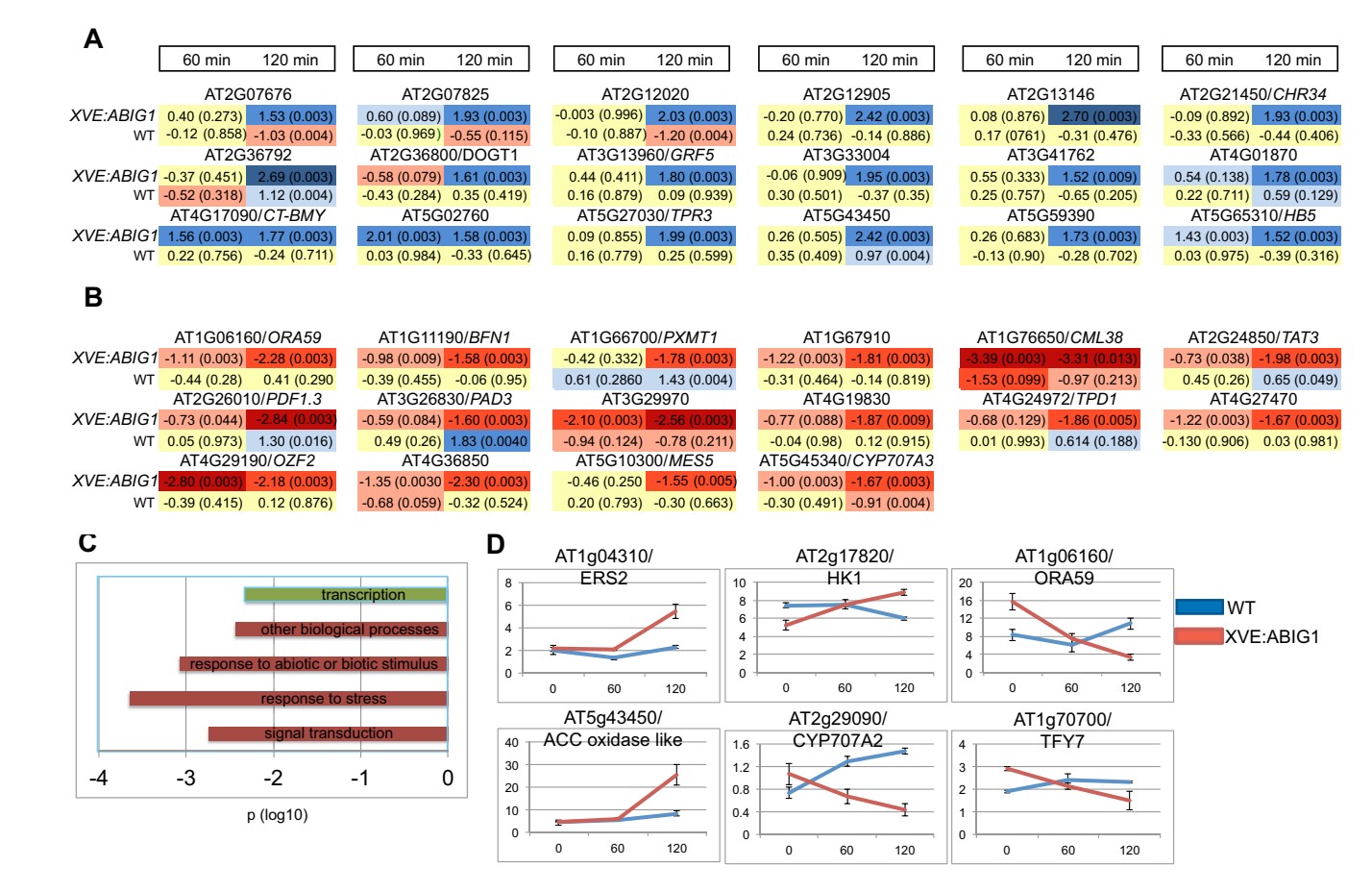

**Figure 6.** Genes regulated in response to ABIG1 induction by estradiol. (**A**) Genes up-regulated in response to *ABIG1* induction by estradiol. (**B**) Genes down-regulated in response to *ABIG1* induction by estradiol. Fold change relative to time 0 is given in log base 2. Darker blue indicates increased fold change. Darker red indicates decreased fold change. The number in parenthesis is p value adjusted for multiple hypothesis testing. Time is given in minutes following estradiol addition. (**C**) Enrichment in gene ontology terms for genes up-regulated (green) and down-regulated (red) by induction of *ABIG1* with estradiol. X axis indicates the level of significance (log 10 p) for each set of terms. (**D**) Expression of ethylene (left), ABA (middle) and jasmonate (right) pathway genes in response to *ABIG1* induction with estrogen.

The following figure supplement is available for figure 6:

**Figure supplement 1.** mRNA levels of cytokinin biosynthetic gene *CYP735A2/At1g67110* in *abig1* mutants and in response to exogenously added ABA.

since these hormone pathways are known to cross-regulate one another. It is therefore unclear which of these genes are primary, direct targets of ABIG1 binding and which are secondary, indirect targets.

*SAG113/HAI1/AT5g59220* encodes a PP2C phosphatase homologous to those in the core ABA signaling pathway. Its transcription is increased rapidly both in response to senescence and in response to ABA (*Zhang and Gan, 2012*). As such, it is a good candidate for regulation by *ABIG1*. However, the transcript corresponding to this locus did not show regulation by induced *ABIG1* in the above experiment (*Figure 6*). We also did not observe a requirement for *ABIG1* in ABA mediated up-regulation of *SAG113/HAI1* (*Figure 5— figure supplement 2*). In addition, *ABIG1* transcript induction did not alter levels of AtNAP (AT1G69490; *Zhang and Gan, 2012*) , a known positive regulator of *SAG113/HAI1* through which ABA is thought to act. Thus, ABA up-regulation of *SAG113/ HAI1*likely occurs through a parallel pathway. Such a parallel pathway may be primarily responsible

for up-regulation of rapidly regulated targets while *ABIG* may be primarily involved in regulation of down-regulated genes

Alterations in cytokinin metabolism, particularly increases in cytokinin activity, are associated with both drought tolerance and with inhibition of senescence (*Peleg et al., 2011*; *Kuppu et al., 2013*). While we found no evidence for altered transcription of cytokinin pathway genes in response to transiently induced *ABIG1* transcription (above), we did find that *abig1-1* mutant seedlings have higher steady state levels of mRNA from the *CYP735A2*/At1G67110 cytokinin biosynthetic gene (*Figure 6—figure supplement 1*). This change in steady state values is likely an indirect effect of reduced *ABIG1* action since no change in *CYP735A2* levels were observed within the first two hours of *ABIG1* induction. Interestingly, while the levels of *CYP735A2* remained steady in wild type in response to exogenous ABA, elevated *CYP735A2* mRNA levels decreased in the *abig1* mutant in response to exogenous ABA (*Figure 6—figure supplement 1*). Thus, not only do some ABA responses remain intact in *abig1* mutants, the physiological and developmental context of the *abig1* mutant also results in novel responses to ABA.

## Discussion

### Evidence that *HAT22/ABIG1* is part of an ABA signaling pathway

In this paper, we present six lines of evidence that *ABIG1/HAT22* plays a part in ABA signaling in the regulation of plant growth: 1/ *ABIG1/HAT22* has similar regulation patterns to two other known ABA signaling genes, *PYL6* and *CIPK12*; 2/ Expression of *ABIG1/HAT22* increases in response to exogenous ABA and drought; 3/ This response is dependent on *ABI1* function; 4/ Induction of the *ABIG1/HAT22* transcription factor mimics the application of exogenous ABA; 5/ Loss of function mutations in *ABIG1/HAT22* cause resistance to ABA mediated growth inhibition and to drought induced leaf yellowing; 6/ Induction of *ABIG1/HAT22* regulates CYP707A3, a gene encoding an enzyme that mediates ABA breakdown. A role for *ABIG1/HAT22* in mediating ABA signaled growth inhibition and leaf senescence does not necessarily mean that this effect is direct nor that the pathway does not involve additional plant growth hormones. Ethylene, in particular, has been tightly linked to ABA signaling. Abscisic acid was first discovered for, and is in fact named for, its ability to promote organ abscission, a process known to be regulated by ethylene (*Patterson and Bleecker, 2004*). More recent examples of links between ABA signaling and ethylene signaling include: Two ethylene receptors, ETR1 and ETR2, regulate the sensitivity of germination to ABA (*Wilson et al., 2014*); Abscisic acid biosynthetic enzymes are required for ethylene signaling in the inhibition of rice root growth (*Ma et al., 2014*); A screen for ABA insensitive mutants in Arabidopsis root growth resulted in the recovery of mutations in the ethylene signaling pathway as well as in the auxin signaling pathway (*Thole et al., 2014*). The link between ABA signaling and ethylene signaling appears to be ancient as the central signaling kinase of the ethylene signaling pathway, CTR1, mediates both ABA and ethylene signaling in the moss Physcomitrellum (*Yasumura et al., 2015*).

The finding that induction of *ABIG1/HAT22* alters levels of mRNAs for genes encoding components of the ethylene, ABA and jasmonate pathways (all pathways that have been shown to promote senescence) as well as the senescence regulating transcription factor GRF5 suggests that *ABIG1* promotes leaf yellowing and senescence by invoking multiple pathways.

How can the action of ABA as a growth inhibitor in response to drought be reconciled with the reduced growth observed in ABA biosynthesis mutants? This is especially puzzling since reduced growth in both circumstances - exogenous addition of ABA and reduced ABA due to a mutation in a biosynthetic mutation - have been shown to be mediated in part through the ethylene pathway (e.g. *Sharp and Noble, 2002*; *Thole et al., 2014*) It is possible that ABA has varying effects at different doses. But it is also possible that the phenotypes caused by loss of function mutations in ABA biosynthesis genes are misleading. For one thing, the phenotypes of these mutants reflect the cumulative effect of secondary, tertiary and higher order consequences of ABA deficiency throughout the history of the plant. For another, loss of function in a step of a biosynthetic pathway may cause accumulation of precursors and/or increased flow through side branches of the biosynthetic pathway. The latter appears to be the case for an earlier step in the ABA biosynthetic pathway (*Avendaño-Vázqueza, 2014*) and for some mutants in the auxin biosynthesis pathway (*Brumos et al., 2014*). Thus, the phenotypes of hormone biosynthetic mutants are more correctly considered as reflecting

the cumulative (over development) consequence of missing or reduced enzyme activity rather than reflecting solely the reduction in the levels of the particular hormone in question. As such, the phenotype of biosynthetic mutants may not accurately reflect the role of environmentally induced hormone production in an otherwise normal plant.

## The role of Class II HD-ZIP proteins in the plant

Class II HD-ZIP proteins are transcription factors that have been shown to regulate shoot growth and development (*Turchi et al., 2013*; *Park et al., 2013*; *Rice et al., 2014*). They include a DNA binding homeodomain and a leucine zipper dimerization domain. Class II HD-ZIP proteins bind to a palindromic sequence as dimers and in several cases have been shown to repress their own transcription (*Ohgishi et al., 2001*; *Ciarbelli et al., 2008*). Several members of this family, including the ABIG1/HAT22 protein include an EAR like domain which can interact with TOPLESS family co-repressors to mediate transcriptional inhibition.

Many of the genes within the HD-ZIPII family of genes, like *ABIG1/HAT22,* are regulated by one or both *REVOLUTA* and *KANADI* transcription factors (*Reinhart et al., 2013*; *Huang et al., 2014*; *Brandt et al., 2012*). Three of these, ATHB4, ATHB2 and HAT3 are required for normal leaf development and blade growth (*Turchi et al., 2013*). The expression of two of them, ATHB4 and HAT3, is most prominent in the adaxial leaf domain where REVOLUTA is active and KANADI is excluded. Whether they are involved in the establishment of organ polarity or whether they are the executors of growth control in response to ad/abaxial cues in the leaf is unclear.

The localization of *ABIG1/HAT22* expression primarily to the vascular parenchyma associated with the tracheary elements - the adaxial portion of the vascular strand - may rely on regulation by REVOLUTA (positive) and KANADI (negative) thus explaining the regulation by these developmental regulators.

In addition to regulation by developmental factors, mRNA levels for several of the class II HD-ZIP genes are also regulated by environmental and hormonal inputs, including ABA, auxin and red/far red light with the exact response to stimulus dependent on the particular Class II HD-ZIP gene in question (*Ciarbelli et al., 2008*). It seems likely that a general role for HD-ZIPII proteins is to link environmental and tissue specific signals to growth control.

Two of the ten Class II HD-ZIP genes in Arabidopsis are the distantly related alpha type Class II HD-ZIP genes *ATHB17* and *ATHB18.* Like *ABIG1/HAT22, ATHB17* mRNA is also increased by ABA and by salt and drought stress (*Park et al., 2013*). Also like *ABIG1/HAT22, ATHB17* activity inhibits growth and *athb17* mutant seedlings are resistant to ABA mediated growth inhibition. However, unlike the case for *ABIG1/HAT22, athb17* loss of function mutant seedlings were more drought resistant and *ATHB17* overexpressors were more drought sensitive. Another difference between *ABIG1/HAT22* and *ATHB17* is that *Hymus et al. (2013)* found that *ATHB17* overexpressing plants led to greener, more chloroplast dense cells while we find that overexpressing *ABIG1/HAT22* causes leaves to be yellow and less green. Thus, these distantly related class II HD-ZIPs may share some functions (growth control, regulation by drought) with *ABIG1/HAT22* but differ in others (leaf yellowing versus greening).

Regulation of plant growth through manipulation of class II HD-ZIP factors may lead to crop plants with improved growth characteristics. Recently, transgenic maize lines overexpressing a truncated form of the *Arabidopsis ATHB17* gene, a type alpha class II HD-ZIP gene, lacking a presumed transcriptional repression domain and presumed to act as a dominant negative (*Rice et al., 2014*), have been permitted for crop use by USDA (Monsanto Petition (14-213-01p) for Determination of Non-regulated Status for Increased Ear Biomass MON 874Ø3 Maize.) This is one of a very few GMO traits approved for crop use. In this application, the ATHB17 overexpressing plants are described as having greater ear biomass. This is presumed to result from inhibition of other class II HDZIP proteins by the truncated ATHB17 protein, thus preventing them from inhibiting growth.

Another way to improve growth of crop plants, in particular in response to drought stress conditions, may be to generate simple loss of function mutations in drought regulated class II HD-ZIP genes such as *ABIG1* and thereby reduce growth inhibition in response to environmental stress. Such an approach does not require the introduction of a transgene and may therefore be acceptable to a broader range of consumers.

## Proposed pathway

We propose that drought acts through *ABIG1* to inhibit growth and to accelerate senescence. Drought has been shown to increase levels of the ABA biosynthetic enzyme, NCED9, in the parenchyma cells surrounding the xylem (*Endo et al. 2008*). We propose that increased ABA in the vascular parenchyma acts through the ABI1 ABA co-receptor to increase *ABIG1* mRNA levels. Increased levels of ABIG1 transcription factor in turn regulate a suite of genes, through direct or indirect action, that includes those shown in *Figure 6*. These targets, many of which are previously implicated in stress response and hormone signaling, in turn trigger reduction of growth at shoot apices and hasten leaf yellowing and evocation of the senescence pathway in leaves.

The action of ABA to increase *ABIG1* mRNA levels need not be direct – in fact we did not find any ABRE elements upstream of the *ABIG1* gene. Nevertheless, Q-RTPCR analysis shows that regulation occurs within hours of ABA application and gene trap analysis shows that *ABIG1* expression is strengthened and expanded in the vasculature in response to ABA. Thus, there is potential for a close temporal and spatial linkage of increased ABA and increased *ABIG1* mRNA levels. Our experiments also show that increased ABIG1 levels have profound consequences on the plant, resulting in growth inhibition and leaf yellowing. While it is possible for ABA to formally act through *ABIG1* in a manner that does not require up-regulation of the *ABIG1* transcript, the experimental observations point to ABA up-regulation of ABIG1 transcript, through an as yet and possible indirect pathway, as a simple mechanism through which ABIG1 action is modulated by ABA. Furthermore, the finding that *ABIG1* can down-regulate a gene encoding an ABA catabolic enzyme suggests that ABA can act to increase its own levels via a positive feedback loop that includes *ABIG1/HAT22.*

An unexpected finding of this work is the discovery of mutations, i.e. loss of function *abig1* mutations, that are resistant to ABA and also resistant to drought. To date ABA resistance has been correlated with drought sensitivity presumably due to the inability of stomata to close in mutants compromised in ABA signaling (e.g. *Wang et al., 2009*; *Park et al., 2013*). The normal germination and stomatal closure in *abig1-1* mutants indicates that the branches of the ABA regulatory pathways controlling these processes are intact in the *abig1*-1 mutant. We propose that *ABIG1* participates in a parallel path in which ABA regulates growth and the decision to undergo senescence (*Figure 7*). We note that an increase in ABA levels precedes an increase in *ABIG1/HAT22* levels in maturing and senescing leaves as found in transcriptome inventories of developing Arabidopsis leaves (*Breeze et al., 2011*).

In crop plants such as sorghum and rice, drought conditions often induce premature senescence. Varieties that resist this induced senescence are referred to as possessing a 'stay green' phenotype (*Thomas and Ougham, 2014*). Stay green traits from wild relatives have been incorporated into a variety of crop plants to breed drought resistant varieties. The *abig1-1* mutant behaves like a stay green mutant in its ability to remain green and to maintain an upright shoot under drought conditions. Screens for additional mutants resistant to ABA during plant growth, as opposed to screens for resistance at the germination stage, should be a productive means of selecting novel mutants with stay green phenotypes that can be used to inform breeding of new drought resistant plant varieties.

## Materials and methods

### Plant growth

Soil grown plants were grown in 4 inch pots in ProMix PGX soil mix. Osmocote slow release fertilizer was added to flats as directed by manufacturer. Greenhouses were kept between 20 and 25 degrees C and plants were supplemented with artificial light to achieve 16 hr day length.

Plants grown in sterile media were grown under cool white fluorescent lamps (24 hr light). Sterile medium consisted of 4.3 grams MS Medium and 0.5 g MES buffer per liter and was adjusted to pH 6.5 with 1 M KOH. For solid media, 8 grams phytagar was added per liter. If sucrose was added to the medium, it was added to 0.1% (10 grams per liter).

For experiments in which plants were scored for their ability to grow on ABA containing medium, seedlings were germinated on MS medium (no sucrose) and allowed to grow for 5 days after which they were transferred to MS medium (no sucrose) to which ABA was added to varying concentrations. Phenotypic characterization was carried out two weeks after transfer.

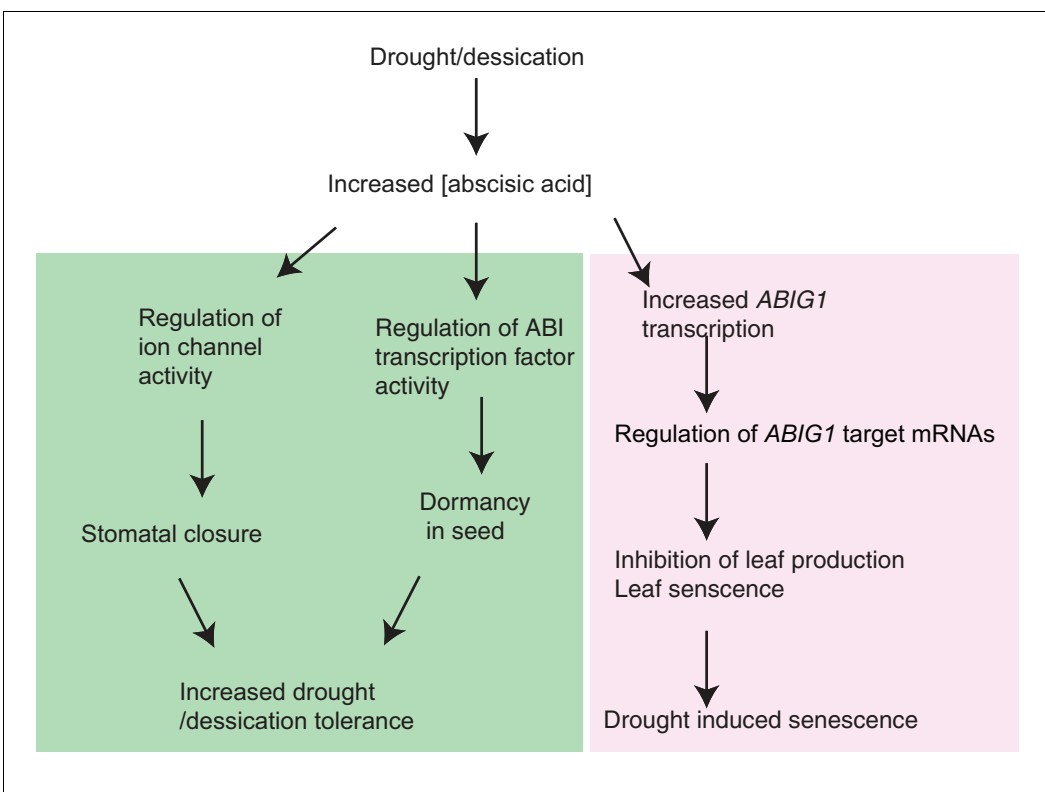

**Figure 7.** Model for *ABIG1* action in the plant. ABA concentration increases following drought. ABIG1 mRNA increases in response and, in a pathway that is distinct from the pathways controlling germination and stomatal closure, causes a reduction in growth and triggers early senescence. In the absence of ABIG1, plant growth is less affected by drought and leaf senescence is less likely to be triggered by drought.

## Morphological characterization

For measurement of morphological characters, plants were photographed and NIH Image J with plugin program LeafJ (*Maloof, 2013*) was used to measure petioles (leaf three), root length and rosette diameter. At least twelve plants were used for each data point.

## Germination assay

For germination assays, seeds were surface sterilized and plated on solid medium containing 5 microMolar ABA (no sucrose) or mock solution. Seeds were stratified at 4°C for two days in the dark before placing plates in the growth chamber (24 hr cool white fluorescent light, 22° C). Seven days later, the fraction of seeds that had germinated was determined under the dissecting microscope. At least 70 seeds were tested per genotype and condition.

## Stomatal closure assay

To measure stomatal closure in response to exogenous ABA, the procedure from *Tseng and Briggs (2010)* was followed. Leaf five (counting the first formed leaf as leaf number 1) was removed from 35-day-old wild type (Ler) and *abig1-1* mutant plants grown in soil under well-watered conditions at 22°C. Strips of epidermis near the midvein were taken from the abaxial side (underside) of leaves using tweezers. The peels were floated on MES-KCl solution (10 mM MES, 50 mM KCl, pH 6.2) under fluorescent lights for 2 hr at 22°C. At that point, half of the samples were treated with MES-KCl-ABA (final concentration of ABA was 5μM) while the other half were treated with MES-KCl-Mock. After two additional hours under lights at 22°C, the leaf strips were immediately dipped into 95% ethanol and mounted onto slides for photography. Images were taken using a compound microscope equipped with Nomarski optics and a 20X objective. The width and length of the

stomatal aperture of 30–40 guard cells per sample were measured using Fiji software. Three biological replicates were used for each data point.

## Genetic stocks

Seeds of *abig1-1* (GT7363, Landsberg erecta background) were obtained from the Cold Spring Harbor Genetrap collection (*Springer, 2000*). Seeds of abig1-4 (SALK_076615, Colombia background) were ordered from the ABRC seed stock center. The line was backcrossed to wild-type plants. F2 plants homozygous for *abig1-1* were isolated in the F2 using PCR to identify the DS insertion. PCR primers used for genotyping are listed in *Supplementary file 1*. The *abi1-1*, *abi2-1* and *abi3-1* mutations were in the Ler background while *abi4-1* was in the Colombia background (*Koorneef et al., 1984*; *Finkelstein, 1994*).

## Plasmid construction and plant transformation

The coding sequence of *ABIG1* (At4g37790) cloned into pENTR223 was obtained from the ABRC in the clone GC105403. The *ABIG1* coding sequence was recombined into the pMDC7 vector to generate a construct in which the *ABIG1* coding sequence was placed under the control of a promoter responsive to the XVE estradiol-inducible transcriptional activator (*Zuo et al., 2000*). This plasmid, named pTL1, was transformed into Agrobacterium strain GV3101 and transformed into wild type Col-0 plants using the floral dip method (*Clough and Bent, 1998*). Drug resistant, transformed plants were selected and then screened for their ability to overexpress *ABIG1* when treated with estradiol.

## Histological analysis

GUS staining was visualized in thin sections of *abig1-1/+* plants using a protocol modified from *Donnelly et al. (1999)*. A leaf piece from individual plants was used to genotype plants at the *ABIG1*-1 locus using PCR with allele specific primers (*Supplementary file 1*). Heterozygous *abig1-1/+* seedlings were prefixed in 90% acetone on ice for 20 min before rinsing in GUS buffer containing 3 mM ferricyanide and 75 µg/ml X-Gluc. Samples in GUS buffer were placed under gentle vacuum with lids off for 8 hr at room temperature and then washed in 70% ethanol four times for 15 min each time. For thin sectioning, after GUS staining, plants were fixed in GAP fixative buffer (3% glutaraldehyde in 25 mM phosphate buffer with 1.6% paraformaldehyde) overnight. The samples were postfixed in 0.5% ruthenium tetroxide in 25 mM phosphate buffer for an hour. The plants were rinsed in water 3 times and dehydrated in an ethanol gradient (15%, 30%, 40%, 50%, 60%, 70%, 85%, 95%, 100%; 10 min for each step) and 100% acetone before exchanging the ethanol into a Spur's resin/acetone gradient solution (1:3, 1:1, 3:1) and then incubated overnight. The plants were transferred and embedded in molds containing 20 ul of prepolymerized embedding medium and then cured at 55°C for 72 hr. The specimen blocks were trimmed and sectioned with glass knives using a Leica MS 5 microtome. The sections were stained in 0.01% aqueous safranin for 15 s and washed twice with DI water then mounted in Cytoseal mounting medium before photographing.

## RNA sequencing and data analysis

Wild-type Col-0 and *XVE:ABIG1* seeds (50 per flask) were surface sterilized, cold treated at 4°C for 48 hr and added to 100 mls of Murashige and Skoog liquid medium with 1% sucrose (pH 6.5) in a 250 ml ehrlenmeyer flask. Flasks were shaken gently at 20°C in constant light for 10 days. At 10 days, β-estradiol was added to 5 microMolar and plants were returned to the shaker. Plants were harvested after 0, 60, and 120 min of treatment. Plants were blotted and flash frozen in liquid nitrogen. Each time point and genotype was replicated three times except the 120 min time point for XVE:ABIG1 which was replicated twice. Total RNA was isolated using the Macherey-Nagel Nucleo-Spin RNA isolation kit according to the manufacturer (Clontech). Total RNA was tested using an Agilent Bioanalyzer. Samples with RIN (RNA integrity number) over 8 were subjected to Illumina sequencing.

Oligo-dT primed libraries were made from the resulting total RNA and sequenced by Otogenetics (Norcross, Georgia) using Illumina technology. Fragment counts were normalized via the median of the geometric means of fragment counts across all libraries, as described in *Anders and Huber (2010)*. The Cuffdiff program was run on the normalized data to identify

expression differences between time 0 and the 60 and 120 min time points (*Trapnell et al., 2013*). P-values were adjusted for multiple hypothesis testing using the Benjamini-Hochberg method (*Benjamini and hochberg, 1995*).

Data are stored on the Gene Expression Omnibus server (Series GSE70100).

### Real-time PCR

RNA was extracted using RNeasy kits from Qiagen. First-strand cDNA was made from 2 micrograms of RNA using the Tetro cDNA synthesis kit and samples were DNase treated according to manufacturer's instructions (Bioline). cDNA was diluted 1 to 5 into RNAse free water to a total of 100 µl and 2 µl of the diluted cDNA was used to perform qRT-PCR. PCR was done using gene-specific primers (*Supplementary file 1*) in technical triplicates on a LightCycler 480 system using the Sensifast SYBR Master mix (Bioline). The ratio of experimental target mRNA to an actin control for each sample was calculated by Applied Biosystems software. An average and standard error of the mean for the three biological replicates and standard deviation were calculated in Excel.

### Drought treatment

Plants were grown in soil in film-covered pots with two holes in the film for plants. One wild type and one *abig1-1* mutant plant were grown per pot. Each experiment was initiated with 18 pots containing equal amounts of soil. If any of the plants didn't germinate well, the entire pot was discarded. Water was withheld from half the pots beginning on day 35. The water content of the pots was monitored by weighing the pots every other day for two weeks beginning with day 0 – the first day with no water. The number of yellow leaves and branches were recorded after 17 days of withheld water. This experiment was repeated three times for *abig1-1* and twice for *abig1–4*.

The chlorophyll content was measured, with minor modifications, according to the method of *Hensel et al. 1993*. Chlorophyll measurement was examined every other day for 12 days beginning with day 1 (first day of withholding water). Each data point represents the average of 10 individual plants (abig1-1/Ler). Samples were taken and pooled from the leaf tip region from leaf 3–8. Samples used for total chlorophyll measurements were ground in liquid nitrogen and then suspended in 1 mL of 80% acetone and quantified photometrically in Tecan Safire plate reader and calculated using the method of *Arnon (1949)*.

### Experimental repeats, replicates and statistical analysis

For gene expression experiments (RNA seq and qRTPCR) three biological replicates were included for each data point. A biological replicate is a flask of 50 seedlings treated independently. For qRTPCR three to four technical replicates were done. For RNA seq, no technical replicates were done. Graphs show averages plus SEMs for the biological replicates.

Sample sizes (i.e. number of biological replicates) of three were chosen for gene expression experiments to allow for a measure of intersample variability while still holding costs down.

For experiments in which plant phenotypes were scored, experiments were repeated three times (e.g. stomatal closure assay, seed germination assays, vegetative growth response to ABA, overexpression of ABIG1, drought response of *abig1-1*mutants). Sample sizes (i.e. number of plants scored) were chosen based on our ability to grow and process a reasonable number of plants.

Data were compared using a 2-way ANOVA test or a t-test as indicated in the text for the relevant experiment.

## Acknowledgements

We thank Cold Spring Harbor Laboratories for the gift of the GT7363 line. We thank Matthew Evans and Virginia Walbot for critical evaluation of the manuscript. Data are stored on the Gene Expression Omnibus server (Series GSE70100). The work in this manuscript was supported by Grant #0929413 from the National Science Foundation to MKB.

## Additional information

### Funding

| Funder | Grant reference number | Author |
|---|---|---|
| National Science Foundation | #0929413 | M Kathryn Barton |
| Carnegie Institution for Science | Endowment | M Kathryn Barton |

The funders had no role in study design, data collection and interpretation, or the decision to submit the work for publication.

### Author contributions

TL, MKB, Conception and design, Acquisition of data, Analysis and interpretation of data, Drafting or revising the article; ADL, FT-R, Conception and design, Acquisition of data, Analysis and interpretation of data; SAH, Conception and design, Analysis and interpretation of data

### Author ORCIDs

M Kathryn Barton, http://orcid.org/0000-0002-5516-1835

## Additional files

### Supplementary files

• Supplementary file 1. RNA expression data tables. (A) RNA sequence data for genes showing up-regulation in response to estradiol induction of XVE:ABIG1 plants. (B) RNA sequence data for genes showing down-regulation in response to estradiol induction of XVE:ABIG1 plants. (C) Expression of cytokinin network genes in response to estradiol induced activation of ABIG1. (D) Expression of ethylene network genes in response to estradiol induced activation of ABIG1. (E) Expression of abscisic acid network genes in response to estradiol induced activation of ABIG1. (F) Expression of chlorophyll degradation network genes in response to estradiol induced activation of ABIG1. (G) Expression of jasmonate network genes in response to estradiol induced activation of ABIG1. (H) Sequence of primers used for genotyping and for Q-RT-PCR.

### Major datasets

The following dataset was generated:

| Author(s) | Year | Dataset title | Dataset URL | Database, license, and accessibility information |
|---|---|---|---|---|
| Barton MK, Hokin S, Liu T, Talavera-Rauh F | 2015 | Identification of mRNAs Regulated in Response to Transcriptional Activation of the Arabidopsis Abscisic Acid Insensitive Growth 1 (AIG1) Transcription Factor cDNA | https://www.ncbi.nlm.nih.gov/geo/query/acc.cgi?acc=GSE70100 | Publicly available at the NCBI Gene Expression Omnibus (accession no: GSE70100) |

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
