## [Decision Letter]

Thank you for submitting your article "The Arabidopsis transcription factor *ABIG1* relays ABA signaled growth inhibition and drought induced senescence" for consideration by *eLife*. Your article has been favorably evaluated by Detlef Weigel as the Senior editor and two reviewers, including Rick Amasino who is a member of our Board of Reviewing Editors.

The reviewers have discussed the reviews with one another and the Reviewing Editor has drafted this decision to help you prepare a revised submission.

Summary:

The insensitivity of the *abig* mutant to ABA-mediated growth inhibition and senescence, but not to other classic ABA effects on seed germination, guard cell closure, and inhibition of primary root growth is an interesting finding. Your work indicates that *ABIG1* appears to define a branch of the ABA pathway that regulates growth and leaf senescence. That the phenotype of inducible overexpression is "opposite" to that of the mutant lends support to the role of *ABIG* in ABA-mediated growth inhibition and senescence, although a minor point is that the "opposite phenotype" is a "sickly" plant, and overexpression of many proteins might cause this.

Essential revisions:

1) One issue is the extent to which *ABIG* is in fact regulated by the core ABA response pathway. There is not a strong effect of *abi1* or ABA. Of course *ABIG* could be an important component of a branch of the ABA pathway that regulates growth and leaf senescence even if it is not itself regulated strongly at the level of mRNA abundance by the ABA pathway. This issue could be more clearly defined in the paper.

2) The leaf senescence phenotype could be presented more rigorously. For example, in the section "*ABIG1/HAT22* mutants show less leaf yellowing in response to drought" there is only a visual assessment of leaf yellowing. A visual assessment is subjective, and also the degree of leaf senescence based upon visual observation can be obscured by differences in anthocyanin accumulation. This is particularly important to consider in the descriptions of leaf senescence in whole plants. A paper by Hensel et al. (Plant Cell. 1993. 5:553-64) provides an example of a simple assay of chlorophyll content as a more quantitative measure of leaf senescence. Also, for Figure 5 (percentage of leaves that had turned yellow) there is no mention of the number of plants or leaves analyzed in the figure legend or in Methods. For the lodging phenotype it is noted that "mutant plants remained erect." Although it is not necessary to create another figure for this, perhaps a bit more could be said in the text about the range of this phenotype.

3) As noted above, the yellowing, wilting, and growth phenotypes indicate there is a specific branch in the ABA response pathway that regulates these things, but the molecular characterization by transcriptomics is not very specific. A great deal has been published about senescence and its associated molecular events. For example, many senescence-induced genes are ABA-regulated at the transcriptional level quite quickly after ABA application (before yellowing occurs). Are the *abig1* mutants defective in ABA-mediated regulation of such senescence-associated transcripts? We recognize that the transcriptomics using the inducible XVE construct addresses this issue to some extent, but more specifically evaluating the role of *ABIG1* in expression of genes known to be involved of ABA-controlled responses would better define this branch of the pathway. For example, SAG113 is an ABA/senescence regulated PP2C (a.k.a. HAI1 (highly aba induced 1) and some NAC transcription factors are reported to be induced during and involved in ABA signaling during senescence.

4) The authors note that selectively blocking ABA-mediated senescence might be an efficient route to improved stress tolerance and promote a desirable stay-green phenotype. The authors should discuss this idea in the context of cytokinin as well, because the same phenotypic outcome has been engineered by using increasing cytokinin content under drought (e.g., see work from Blumwald and colleagues).

*Reviewer #1:*

The authors conclusion that "*ABIG1* function is required for ABA-mediated inhibition of leaf production and petiole growth" appears to be well substantiated by results – for example, in Figure 3, the difference in the 5 μm ABA effect in wt versus the *abig1* mutant is quite striking. In Figure 4, the inducible overexpression phenotype is quite striking as well, although I suspect that many proteins if sufficiently overexpressed will "muck things up" and result in a striking phenotype.

The weakest part of the paper is the section entitled "*ABIG1/HAT22* mutants show less leaf yellowing in response to drought." Yellowing is a very subjective thing and the degree of leaf senescence based upon visual observation can be obscured by differences in anthocyanin accumulation (i.e., impaired anthocyanin accumulation can be mistaken for more rapid senescence). For Figure 5 (percentage of leaves that had turned yellow) there is no mention of the number of plants or leaves analyzed in the figure legend or in Methods.

Also in this section are other phenotypes like "mutant plants remained erect." One trusts that this is highly reproducible, but that ought to be discussed.

In the Discussion, the Monsanto stuff is not "tied in" well to the theme of the paper. "In this application, the ATHB17 overexpressing plants are described as having greater ear biomass. This is presumed to result from inhibition of other class II HDZIP proteins by the truncated ATHB17 protein."

*Reviewer #2:*

Liu et al. report on the characterization of *ABIG1* and its role in ABA-mediated senescence and growth inhibition. *ABIG1* was identified as a transcript that is oppositely regulated by Kanadi and Revoluta, a class of genes the authors have previously defined as ORKs. The link between *ABIG1* and ABA was made because ORKs are enriched in genes involved in ABA responses. *ABIG1* has previously been described as the homeodomain-containing gene *HAT22*. Prior data has shown that *HAT22*'s mRNA levels are regulated by water stress, and that phenotype is recapitulated in this work and attributed, at least in part, to regulation by the core ABA-response pathway.

In comparison to previously reported mutants that affect ABA signaling, the novelty reported in this paper is that *abig1* mutants are less sensitive to ABA-mediated growth inhibition and senescence than wild type, but do not display defects in other classic ABA responses such as seed germination, guard cell closure and inhibition of primary root growth. Thus, *abig1* appears to define a branch of the ABA pathway that regulates growth and leaf senescence, two under studied but important components of ABA action. Over-expression of *ABIG1* using a XVE-inducible form yields expected phenotypes that are opposite to those observed in the loss-of-function mutant alleles. Using a water deprivation experiment the authors make the observation that *abig1* loss-of-function mutants are more drought tolerant, as suggested by increased shoot and root growth, reduced leaf senescence, and reduced lodging. This observation suggests that selectively blocking ABA-mediated senescence might be an efficient route to improved stress tolerance and inter-related stay-green phenotypes. The authors should probably discuss this idea in the context of cytokinin as well, because the same phenotypic outcome has been engineered by using increasing cytokinin content under drought (i.e. see work from Blumwald and colleagues).

The most critical point of the work is that it implies a new branch in the ABA response pathway that regulates both senescence and growth, but the molecular characterization of both these processes is quite cursory. A great deal has been written about senescence and its associated molecular events, but all we can say from the work presented is that the leaves in the *abig1* mutants don't yellow as much. For example, many senescence-induced genes are ABA-regulated at the transcriptional level quite quickly after ABA application and before yellowing occurs. Are the *abig1* mutants defective in ABA-mediated regulation of senescence-associated transcripts? I recognize that the inducible XVE constructs go somewhat towards addressing this issue, but a clearer delineation of the necessity for *ABIG1* in mediating a subset of ABA-controlled responses would add to the work substantially. The direct link between ABA and *ABIG1* hinges on an increase in *ABIG1* transcript levels that is only partially affected by the *abi1-1* mutation. How direct is the regulation?

Overall, I believe this is a potentially important and pioneering paper, but it would benefit from greater depth in its physiological analyses and by building a stronger case for a direct molecular link between components of the ABA signaling network and *ABIG1*.

---

## [Author Response]

Essential revisions:

*1) One issue is the extent to which ABIG is in fact regulated by the core ABA response pathway. There is not a strong effect of abi1 or ABA. Of course ABIG could be an important component of a branch of the ABA pathway that regulates growth and leaf senescence even if it is not itself regulated strongly at the level of mRNA abundance by the ABA pathway. This issue could be more clearly defined in the paper.*

The reviewers point out that the changes in *ABIG1* mRNA levels in response to ABA are relatively small and that, while ABIG could still act downstream of ABA, the alteration in transcript levels in response to ABA may not be critical.

We agree and have addressed this criticism by including a paragraph in the Discussion (subsection “Proposed pathway”) that presents our argument for why transcriptional up-regulation of *ABIG1/HAT22* (albeit likely indirect) is the simplest mechanism of interaction between ABA and *ABIG1* for the model. We’ve also rearranged the section on *abig1/+* enhancer trap expression in hypocotyls in response to ABA which shows more clearly the increase in strength and extent of expression in response to ABA.

*2) The leaf senescence phenotype could be presented more rigorously. For example, in the section "ABIG1/HAT22 mutants show less leaf yellowing in response to drought" there is only a visual assessment of leaf yellowing. A visual assessment is subjective, and also the degree of leaf senescence based upon visual observation can be obscured by differences in anthocyanin accumulation. This is particularly important to consider in the descriptions of leaf senescence in whole plants. A paper by Hensel et al. (Plant Cell. 1993. 5:553-64) provides an example of a simple assay of chlorophyll content as a more quantitative measure of leaf senescence. Also, for Figure 5 (percentage of leaves that had turned yellow) there is no mention of the number of plants or leaves analyzed in the figure legend or in Methods. For the lodging phenotype it is noted that "mutant plants remained erect." Although it is not necessary to create another figure for this, perhaps a bit more could be said in the text about the range of this phenotype.*

The reviewers feel that the senescence phenotype could be presented more rigorously. We have used the method by Hensel et al., suggested by the reviewers to measure chlorophyll levels in wild type and in *abig1* mutant plants in response to drought. This experiment required us to set up a replicate of the drought experiment and has taken the bulk of the time in preparing this revision. The new data are in Figure 5 and largely parallel what we have seen before. We have also added numbers of plants and leaves used in the study.

*3) As noted above, the yellowing, wilting, and growth phenotypes indicate there is a specific branch in the ABA response pathway that regulates these things, but the molecular characterization by transcriptomics is not very specific. A great deal has been published about senescence and its associated molecular events. For example, many senescence-induced genes are ABA-regulated at the transcriptional level quite quickly after ABA application (before yellowing occurs). Are the abig1 mutants defective in ABA-mediated regulation of such senescence-associated transcripts? We recognize that the transcriptomics using the inducible XVE construct addresses this issue to some extent, but more specifically evaluating the role of ABIG1 in expression of genes known to be involved of ABA-controlled responses would better define this branch of the pathway. For example, SAG113 is an ABA/senescence regulated PP2C (a.k.a. HAI1 (highly aba induced 1) and some NAC transcription factors are reported to be induced during and involved in ABA signaling during senescence.*

We have tested the *SAG113/HAI1* gene for upregulation by ABA in the *abig1* mutant. Upregulation occurs similar to wild type (Figure 5—figure supplement 2) indicating that for this senescence upregulated gene, *ABIG1* function is dispensable. Given the likely role of *ABIG1* as a transcriptional repressor, it is possible that most ABA/senescence upregulated genes are activated through a parallel pathway.

*4) The authors note that selectively blocking ABA-mediated senescence might be an efficient route to improved stress tolerance and promote a desirable stay-green phenotype. The authors should discuss this idea in the context of cytokinin as well, because the same phenotypic outcome has been engineered by using increasing cytokinin content under drought (e.g., see work from Blumwald and colleagues).*

The reviewers have asked us to discuss cytokinin and its role in drought resistance. We do this and reference papers by Blumwald and coworkers. We also include qRTPCR data showing that steady state levels of a cytokinin biosynthetic enzyme are increased in the mutant (Figure 6—figure supplement 1). These data are discussed in the text.

*Reviewer #1:*

*The authors conclusion that "ABIG1 function is required for ABA-mediated inhibition of leaf production and petiole growth" appears to be well substantiated by results – for example, in Figure 3, the difference in the 5 μm ABA effect in wt versus the abig1 mutant is quite striking. In Figure 4, the inducible overexpression phenotype is quite striking as well, although I suspect that many proteins if sufficiently overexpressed will "muck things up" and result in a striking phenotype.*

It is true that the overexpression of proteins to high levels may well “muck things up” in ways that are rather non-specific. That said, overexpression of Class I (A. Himmelbach et al., 2002, EMBO J. 21:3029), Class III (multiple observations by Barton and colleagues) and Class IV HD-ZIP (Ohashi et al., 2002, Plant Journal 29: 359.) and even more distantly related families of class II HD-ZIP (Park et al., 2013) proteins do not result in the yellowing stunted growth phenotype so this is at least a unique response to overexpression within the HD-ZIP clade of transcription factors.

*The weakest part of the paper is the section entitled "ABIG1/HAT22 mutants show less leaf yellowing in response to drought." Yellowing is a very subjective thing and the degree of leaf senescence based upon visual observation can be obscured by differences in anthocyanin accumulation (i.e., impaired anthocyanin accumulation can be mistaken for more rapid senescence). For Figure 5 (percentage of leaves that had turned yellow) there is no mention of the number of plants or leaves analyzed in the figure legend or in Methods.*

Measurement of chlorophyll in wt and mutant in response to drought is addressed above in Point 2.

*Also in this section are other phenotypes like "mutant plants remained erect." One trusts that this is highly reproducible, but that ought to be discussed.*

Failure of plants to remain erect was unique to the wt plants in these experiments. We recorded numbers for one replicate experiment and these are included in the text.

*In the Discussion, the Monsanto stuff is not "tied in" well to the theme of the paper. "In this application, the ATHB17 overexpressing plants are described as having greater ear biomass. This is presumed to result from inhibition of other class II HDZIP proteins by the truncated ATHB17 protein."*

The section on ATHB17 was altered to indicate the observed role in growth control of members of this gene family and to indicate the potential parallels between loss of function ABIG1 mutants and dominant negative ATHB17 transgenes in allowing extra growth. A reference to published work on the truncated ATHB17 line in maize was added as well as work on the response of the distantly related ATHB17 gene and associated mutants to ABA and stress.

*Reviewer #2:*

*Liu et al. report on the characterization of ABIG1 and its role in ABA-mediated senescence and growth inhibition. ABIG1 was identified as a transcript that is oppositely regulated by Kanadi and Revoluta, a class of genes the authors have previously defined as ORKs. The link between ABIG1 and ABA was made because ORKs are enriched in genes involved in ABA responses. ABIG1 has previously been described as the homeodomain-containing gene HAT22. Prior data has shown that HAT22's mRNA levels are regulated by water stress, and that phenotype is recapitulated in this work and attributed, at least in part, to regulation by the core ABA-response pathway.*

*In comparison to previously reported mutants that affect ABA signaling, the novelty reported in this paper is that abig1 mutants are less sensitive to ABA-mediated growth inhibition and senescence than wild type, but do not display defects in other classic ABA responses such as seed germination, guard cell closure and inhibition of primary root growth. Thus, abig1 appears to define a branch of the ABA pathway that regulates growth and leaf senescence, two under studied but important components of ABA action. Over-expression of ABIG1 using a XVE-inducible form yields expected phenotypes that are opposite to those observed in the loss-of-function mutant alleles. Using a water deprivation experiment the authors make the observation that abig1 loss-of-function mutants are more drought tolerant, as suggested by increased shoot and root growth, reduced leaf senescence, and reduced lodging. This observation suggests that selectively blocking ABA-mediated senescence might be an efficient route to improved stress tolerance and inter-related stay-green phenotypes. The authors should probably discuss this idea in the context of cytokinin as well, because the same phenotypic outcome has been engineered by using increasing cytokinin content under drought (i.e. see work from Blumwald and colleagues).*

See point 4 above for addition of experiments and text addressing the potential role of cytokinin in drought resistance in *abig1* mutant plants.

*The most critical point of the work is that it implies a new branch in the ABA response pathway that regulates both senescence and growth, but the molecular characterization of both these processes is quite cursory. A great deal has been written about senescence and its associated molecular events, but all we can say from the work presented is that the leaves in the abig1 mutants don't yellow as much. For example, many senescence-induced genes are ABA-regulated at the transcriptional level quite quickly after ABA application and before yellowing occurs. Are the abig1 mutants defective in ABA-mediated regulation of senescence-associated transcripts?*

See point 3 above for addition of experiments and text addressing the role of ABIG1 in regulating senescence associated transcripts.

*I recognize that the inducible XVE constructs go somewhat towards addressing this issue, but a clearer delineation of the necessity for ABIG1 in mediating a subset of ABA-controlled responses would add to the work substantially. The direct link between ABA and ABIG1 hinges on an increase in ABIG1 transcript levels that is only partially affected by the abi1-1 mutation. How direct is the regulation?*

*Overall, I believe this is a potentially important and pioneering paper, but it would benefit from greater depth in its physiological analyses and by building a stronger case for a direct molecular link between components of the ABA signaling network and ABIG1.*

See point 1 above for discussion of direct vs. indirect regulation of ABIG1 transcription by ABA.